# Uptake and cardiac events of COVID-19 vaccinations among Canadian youth and young adults

Kimball Zhang[1,2], Emilie Terebessy[1], Jingqin Zhu[1,2], Catherine Birken[2,3], Cornelia M. Borkhoff[3], Andrea Gershon[1,2,4,5], Theo J. Moraes[6], Tetyana Kendzerska[2,7], Smita Pakhale[7], Teresa To[1,2,4] *

1 Child Health Evaluative Sciences, The Hospital for Sick Children, Toronto, Ontario, Canada, 2 ICES, Toronto, Ontario, Canada, 3 Division of Paediatric Medicine and the Paediatric Outcomes Research Team (PORT), The Hospital for Sick Children, Toronto, Ontario, Canada, 4 Dalla Lana School of Public Health, University of Toronto, Toronto, Ontario, Canada, 5 Sunnybrook Health Sciences Centre, Toronto, Ontario, Canada, 6 Division of Respiratory Medicine, The Hospital for Sick Children, Toronto, Ontario, Canada, 7 Department of Medicine, Ottawa Hospital Research Institute, University of Ottawa, Ottawa, Ontario, Canada

* teresa.to@sickkids.ca

**Data Availability Statement:** The dataset from this study is held securely in coded form at ICES. While legal data sharing agreements between ICES and data providers (e.g., healthcare organizations and

## Abstract

Few studies have examined population-level data of the COVID-19 original and bivalent vaccine on its uptake and potential side effects. We used population-based health administrative data from Jan 2021–Feb 2023 to identify Ontario residents aged 12–35 years old to calculate their rates of COVID-19 vaccine uptake and vaccine-related cardiac events (myocarditis and pericarditis). Multivariable Cox, logistic, and negative binomial regression analyses were used to adjust for covariates. Hazard ratios (HR) were reported with 95% confidence intervals (CI). The study population included 5,012,721 individuals. Comparing to the general population, those with chronic diseases were associated with 13–37% higher rates of vaccine uptake and 1.39–2.27 times higher odds of receiving booster doses. Overall, post-vaccination cardiac event incidence rates ranged from 3–12 per 100,000 persons. Compared to the general population, the incidence rate of cardiac events among those with asthma and allergic diseases was significantly higher, 3.7 events per 100,000 persons. Compared to the general population, those with asthma and/or allergic diseases had significantly higher associated likelihoods of a cardiac event (HR = 1.31, 95% CI: 1.08–1.57). Females, adults, and those with prior COVID-19 infections had decreased odds of cardiac events after 2nd vaccine doses. No significant differences in post-vaccine cardiac events were detected between original and bivalent doses. This Canadian population-based study reported substantially higher rates of vaccine uptake and a very rare incidence of temporally associated post-vaccination cardiac events. While substantially smaller than the benefits of vaccination, our results indicated a continued small risk of cardiac side effects from bivalent COVID-19 vaccines in individuals with comorbidities.

government) prohibit ICES from making the dataset publicly available, access may be granted to those who meet pre-specified criteria for confidential access, available at www.ices.on.ca/ DAS (email: das@ices.on.ca). This does not alter our adherence to PLOS ONE policies on sharing data and materials.

**Funding:** This study was funded by the Ontario Ministry of Health (MOH). Dr. Teresa To is funded by a Canadian Institutes of Health Research Tier 1 Canada Research Chair in Asthma. This study was supported by ICES (formerly the Institute for Clinical Evaluative Sciences), which is funded by a grant from the Government of Ontario. This work was also supported by the Ontario Health Data Platform (OHDP), a Province of Ontario initiative to support Ontario's ongoing response to COVID-19 and its related impacts. The analyses, conclusions, opinions, and statements expressed herein are those of the authors, and not necessarily those of the data sources; no endorsement is intended or should be inferred. The funders of this study did not play a role in either study design, data collection, analysis, interpretation of findings, decision to publish or preparation of the manuscript.

**Competing interests:** The authors have declared that no competing interests exist.

## Introduction

Long-term efforts and outreach resulted in 83.8% of Ontario's population being vaccinated with at least one dose of a COVID-19 vaccine, as of June 2023 [1]. However, concerns over the vaccine and its adverse side effects remain, especially for youth. Studies examining individuals after they received the original COVID-19 vaccine found a temporal association between the vaccine and cardiovascular complications, including myocarditis and pericarditis. These complications most commonly manifested in young adults, especially males [2]. There is limited information on whether persons with chronic conditions such as asthma and allergic diseases are more at risk, however it is hypothesized due to the predilection to hypersensitivity [3].

The evolution of the COVID-19 virus from its original form to its dominant subvariants over time, such as Delta, Omicron, and their sub-lineages, coupled with the ongoing development of vaccines to combat said variants, complexify our understanding of vaccine-related cardiovascular complications. Previous studies estimated that COVID-19 related myocarditis occurred among 1–4% of COVID-19 patients [4], but were much higher (27.8%) in severe COVID-19 pneumonia cases [5]. More recent studies of the Omicron-dominant wave also reported higher hospital admissions of poor outcomes, including myocarditis [6]. In contrast, findings from clinical trials of the newly developed bivalent vaccine by Pfizer and Moderna suggested no vaccine-related cases of myocarditis or pericarditis [7], but the limited number of participants in the clinical trials presents challenges in determining/generalizing the risk of a very rare event occurring 1–4 per 100,000 persons [8].

Few population-based studies have been conducted to examine the uptake and side effects of the COVID-19 original and bivalent vaccines. Fewer still have focused on adolescents and young adults with chronic conditions, who may be at a greater risk of COVID-19 and vaccine side effects. While clinical trial information suggested that bivalent vaccines were not associated with risks of myocarditis and pericarditis, this remained to be seen at a larger population scale. Understanding how different factors play into the uptake of the COVID-19 vaccine and its side effects can better inform future patients, healthcare practitioners, and vaccine outreach. Therefore, this study aims to use population-based health administrative data to examine patterns of the uptake of the COVID-19 vaccine and to quantify the association and risks of temporally associated cardiac-related side effects among youth and young adults.

## Methods

### Study design & population

The uptake of vaccination was investigated using a longitudinal cohort design. The study population included all Ontario residents aged 12–35 years as of Jan 1, 2021. Individuals were excluded if they did not have data on age, an Ontario residence code, or a valid Ontario health card number. Individuals were also excluded if they had a history of myocarditis, cardiomyopathy, or pericarditis, were registered as someone with a listed chronic condition, or were diagnosed with a listed chronic condition between 2017–2021. The list of chronic conditions is defined by Ontario's COVID-19 at-risk vaccine guidelines (see S1 Appendix). Asthma, allergic diseases, and diabetes were not excluded.

### Data sources

This study used routinely collected health administrative data for Ontario. In Ontario, there is a publicly funded single-payer healthcare system. Health administrative data were linked using unique encoded identifiers at ICES (formerly the Institute for Clinical Evaluative Sciences). Data on hospital admissions were captured by the Canadian Institute for Health Information

Discharge Abstract Database (CIHI-DAD) while data on emergency department (ED) visits were captured by the National Ambulatory Care Reporting System (NACRS). The Ontario Health Insurance Plan (OHIP) claims database captured outpatient physician office and virtual visits. CIHI-DAD, NACRS, and OHIP data were available from April 1, 1994 to February 28, 2023. The COVID-19 Integrated Testing Data (C19INTGR) captured information relating to laboratory tests and the COVID-19 Vaccination in Ontario (COVaxON) database captured vaccine and recipient information between December 1, 2020 to April 30, 2023. Data on study population characteristics such as age, sex, residence postal code, and immigration status were captured through the Provincial Registered Persons Database and the Immigration, Refugees and Citizenship Canada's Permanent Residents database.

## Exposure & outcome definitions

The primary exposures were diagnosis of asthma, allergic rhinitis, or eczema (AAD) and/or diabetes. Asthma diagnosis was determined based on an administrative case definition of ≥1 hospitalization for asthma, or ≥2 outpatient visits for asthma in two consecutive years. This definition has been previously validated in Ontario with a sensitivity of 84% and a specificity of 77% [9]. Allergic rhinitis (International Classification of Diseases [ICD]-10: J301-J304) and eczema (ICD-10: L20) were defined as any diagnosis of their respective ICD codes.

Primary outcomes were the uptake of COVID-19 vaccine(s) and occurrence of cardiac outcomes. Vaccination status was determined following Ontario's COVID-19 vaccination guidelines [10]. For cardiac outcomes (myocarditis (ICD-10: I40-I41) & pericarditis (ICD-10: I30-I32)), individuals were followed for 14 days after their primary series and first booster vaccine doses (1st, 2nd, and 3rd doses), where applicable, until first diagnosis, if any. Diagnoses from serious events (hospitalisation, ED visit) were included while acute care diagnoses (outpatient) were excluded for primary analyses. Unvaccinated individuals were matched to vaccinated individuals by age group, sex, and Census tract and followed for the same observation period as their matched counterpart. See S2 Appendix for the full list of diagnosis codes.

## Covariates

Regression models were adjusted for potential confounders including age group [12–17, 18–35], sex (male/female), location of residence (rural/urban), census-based income quintiles, socioeconomic status as proxy measured by the Ontario Marginalization Index (ON-Marg) quintiles [11], and recency of immigration (yes/no). Residence was rural if the individual resided in a community with ≤10,000 people, or urban if otherwise true. ON-Marg provided a measure of marginalization at the population-level based on Census information using four dimensions: material deprivation, residential instability, dependency, and ethnic concentration. Based on each participant's residence postal code, they were assigned a score from 1 (least marginalized) to 5 (most marginalized) for each dimension. Immigrants were considered recent if they had immigrated to Canada within the last 10 years (2010 onwards). Cardiac outcome regression models were also adjusted for COVID infection history (yes/no) and type of vaccination doses (unvaccinated, original, mixed). Individuals vaccinated with only non-bivalent vaccines for their first three doses were considered to have original vaccine doses while people with at least one bivalent vaccine were considered to have mixed vaccine doses.

## Statistical analysis

Cox multivariable regression models were used to estimate hazard ratios (HR), while logistic regression models for odds ratios (OR) with 95% confidence intervals (CI) of outcomes. Negative binomial multivariable regression models with a person-time offset were also used to

estimate rate ratios (RR) and 95% CI of outcomes. Individuals deemed ineligible for full/ booster doses (due to age or timing of doses at the time of analysis), or received vaccinations after available health data, were excluded from the respective analyses. Sensitivity analyses including stratification by dose number and acute care diagnoses. All statistical analyses and proportion figures were generated using SAS Enterprise Guide 7.1 (SAS Institute Inc., Cary, NC, USA) and forest plots were created using the *forestplot* package in R statistical computing software version 4.2.2 (https://www.r-project.org/). Chronic conditions for population exclusions were identified by ACG System's Aggregated Diagnosis Groups (version 10). Ethics approval exemption was obtained from the Hospital for Sick Children Research Ethics Board (Toronto, Ontario, Canada).

## Results

### Population characteristics

Table 1 shows that 5,012,721 individuals aged 12–35 years were included in this study. Of these, the majority (80.9%) were aged 18–35 years and 19.1% were aged 12–17 years. The study cohort consisted of 49% females, those largely from areas of low- to middle-income quintiles (62.2%), and the majority (91.6%) resided in urban areas. 17% had prevalent AAD, 0.3% had diabetes only, and 0.8% had both AAD and diabetes. By early 2023, 9.9% had been lab tested-positive for COVID-19, 72.9% of the study cohort had received at least one COVID-19 vaccine dose, and 54,733 (1.1%) received at least one bivalent vaccine in their first three eligible vaccine doses. Out of the total population, six hundred and sixteen (0.01%) individuals had cardiac outcomes; of whom the majority of individuals (96.9%) had cardiac events happen up to 14 days following vaccination. Among those post-vaccination individuals, 505 individuals (84.5%) had cardiac events occur within seven days and two-thirds (66.1%) of cardiac events occurred within two weeks post-2nd dose vaccination (Fig 1). Overall, the observed incidence rates of cardiac events in 0–14 days post-vaccination were 3.2 per 100,000 post-first dose, 11.7 per 100,000 post-second dose, and 4.8 per 100,000 post-third dose.

### Vaccine uptake

Table 2 shows the RRs and 95% CIs of vaccine uptake from negative binomial regressions. Adjusting for confounders, comparing to the general population, individuals with AAD only, diabetes only, and diabetes with AAD were significantly more likely to have any vaccination, be fully vaccinated, or have a booster dose, respectively (RR = 1.13, 95% CI: 1.13–1.13; RR = 1.14, 95% CI: 1.13–1.14; RR = 1.18, 95% CI: 1.17–1.19). Compared to males, females had significantly higher rates of all types of vaccinations along with recent immigrants, compared to non-immigrants, and the younger age group compared to the older, except for booster vaccinations. Fig 2A and 2B shows that at 6-months (182 days), those with only AAD were less likely to be vaccinated in all types compared to those with diabetes only and those with diabetes with AAD, but more likely to be vaccinated than the overall general population. The ORs and 95% CIs of vaccination type from logistic regressions are shown in Fig 1C. Adjusting for covariates, AAD, diabetes, and diabetes with AAD chronic disease groups, compared to the general population, had higher odds of having boosted vaccinations than all other types of vaccination, respectively (OR = 1.39, 95% CI:1.39–1.40; OR = 1.79, 95% CI:1.71–1.89; OR = 2.27, 95% CI: 2.21–2.34), as were females compared to males (OR = 1.22, 95% CI:1.21–1.23). The full results can be found in S1 Table. Youth were higher odds of being fully vaccinated compared to adults (OR = 2.94, 95% CI: 2.92–2.95) and people in lower income quintiles had 9% lower odds of having a boosted vaccination compared to the highest income quintile (OR = 0.91, 95% CI: 0.90–0.92). Recent immigrants had 2-fold higher odds of having any

**Table 1. Select characteristics\* of the study population.**

| Characteristics | | Overall Population | | General Population | | AAD† | | Diabetes | | Diabetes with AAD† | |
|---|---|---|---|---|---|---|---|---|---|---|---|
| | | Number | % | Number | % | Number | % | Number | % | Number | % |
| **Demographic Factors** | | | | | | | | | | | |
| N | | 5,012,721 | | 4,107,464 | 81.9 | 847,549 | 16.9 | 15,313 | 0.3 | 42,395 | 0.8 |
| Age Group | 12–17 | 959,530 | 19.1 | 788,963 | 19.2 | 165,742 | 19.6 | 1,210 | 7.9 | 3,615 | 8.5 |
| | 18–35 | 4,053,191 | 80.9 | 3,318,501 | 80.8 | 681,807 | 80.4 | 14,103 | 92.1 | 38,780 | 91.5 |
| Sex | Female | 2,460,683 | 49.1 | 2,058,438 | 50.1 | 370,264 | 43.7 | 7,706 | 50.3 | 24,275 | 57.3 |
| | Male | 2,552,038 | 50.9 | 2,049,026 | 49.9 | 477,285 | 56.3 | 7,607 | 49.7 | 18,120 | 42.7 |
| Asthma | Yes | 861,568 | 17.2 | NA | - | 847,549 | 100.0 | NA | - | 14,019 | 33.1 |
| | No | 4,151,153 | 82.8 | 4,107,464 | 100.0 | NA | - | 15,313 | 100.0 | 28,446 | 66.9 |
| Residence | Urban | 4,590,548 | 91.6 | 3,767,562 | 91.7 | 770,711 | 90.9 | 13,847 | 90.4 | 38,428 | 90.6 |
| | Rural | 400,095 | 8.0 | 320,859 | 7.8 | 73,989 | 8.7 | 1,427 | 9.3 | 3,820 | 9.0 |
| | Missing | 22,078 | 0.4 | 19,043 | 0.5 | 2,849 | 0.3 | 39 | 0.3 | 147 | 0.4 |
| Recent Immigrant | Yes | 455,621 | 9.1 | 438,838 | 10.7 | 10,179 | 1.2 | 4,296 | 28.1 | 2,308 | 5.4 |
| | No | 4,557,100 | 90.9 | 3,668,626 | 89.3 | 837,370 | 98.8 | 11,017 | 72.0 | 40,087 | 94.6 |
| Prior COVID Infection | Yes | 498,272 | 9.9 | 394,197 | 9.6 | 95,675 | 11.3 | 2,181 | 14.2 | 6,219 | 14.7 |
| | No | 4,514,449 | 90.1 | 3,713,267 | 90.4 | 751,874 | 88.7 | 13,132 | 85.8 | 36,176 | 85.3 |
| Income Quintile | 1 (Lowest) | 1,098,018 | 21.9 | 919,584 | 22.4 | 163,315 | 19.3 | 4,424 | 28.9 | 10,695 | 25.2 |
| | 2 | 1,024,482 | 20.4 | 847,564 | 20.6 | 164,771 | 19.4 | 3,313 | 21.6 | 8,834 | 20.8 |
| | 3 | 998,464 | 19.9 | 815,499 | 19.9 | 171,270 | 20.2 | 3,111 | 20.3 | 8,584 | 20.3 |
| | 4 | 951,942 | 19.0 | 768,405 | 18.7 | 173,455 | 20.5 | 2,452 | 16.0 | 7,630 | 18.0 |
| | 5 (Highest) | 916,057 | 18.3 | 736,122 | 17.9 | 171,500 | 20.2 | 1,964 | 12.8 | 6,471 | 15.3 |
| | Missing | 23,758 | 0.5 | 20,290 | 0.5 | 3,238 | 0.4 | 49 | 0.3 | 181 | 0.4 |
| **Vaccination & Cardiac Events** | | | 0.0 | | 0.0 | | | | | | |
| Vaccination Status | Unvaccinated | 1,359,922 | 27.1 | 1,183,398 | 28.8 | 168,105 | 19.8 | 2,393 | 15.6 | 6,026 | 14.2 |
| | Partial Vaccination | 101,325 | 2.0 | 83,696 | 2.0 | 16,598 | 2.0 | 264 | 1.7 | 767 | 1.8 |
| | Full Vaccination | 1,829,723 | 36.5 | 1,475,545 | 35.9 | 332,895 | 39.3 | 5,836 | 38.1 | 15,447 | 36.4 |
| | Boosted Vaccination | 1,721,751 | 34.4 | 1,364,825 | 33.2 | 329,951 | 38.9 | 6,820 | 44.5 | 20,155 | 47.5 |
| Vaccine Doses Type | Unvacinated | 1,359,922 | 27.1 | 1,183,398 | 28.8 | 168,105 | 19.8 | 2,393 | 15.6 | 6,026 | 14.2 |
| | Original only | 3,598,066 | 71.8 | 2,878,021 | 70.1 | 671,422 | 79.2 | 12,733 | 83.2 | 35,890 | 84.7 |
| | Original-Bivalent Mix | 54,733 | 1.1 | 46,045 | 1.1 | 8,022 | 1.0 | 187 | 1.2 | 479 | 1.1 |
| Days Followed | Median (IQR) | 719 (712–726) | | 712 (712–726) | | 719 (719–726) | | 724 (719–726) | | 719 (719–726) | |
| Days to First Vaccination | Median (IQR) | 21 (8–80) | | 20 (7–79) | | 25 (11–86) | | 24 (10–81) | | 27 (10–99) | |
| Cardiac Event within 14 days | No | 5,012,105 | 100.0 | 4,107,006 | 100.0 | 847,395 | 100.0 | 15307–15312\* | 100.0 | 42389–42394\* | 100.0 |
| | Yes | 616 | 0.0 | 458 | 0.0 | 154 | 0.0 | <6 | 0.0 | <6 | 0.0 |
| | Unvaccinated | 19 | 0.0 | 13 | 0.0 | 6 | 0.0 | 0 | 0.0 | 0 | 0.0 |
| | After dose 1 | 118 | 0.0 | 92 | 0.0 | 22 | 0.0 | <6 | 0.0 | <6 | 0.0 |
| | After dose 2 | 417 | 0.0 | 305 | 0.0 | 112 | 0.0 | 0 | 0.0 | 0 | 0.0 |
| | After dose 3 | 82 | 0.0 | 63 | 0.0 | 19 | 0.0 | 0 | 0.0 | 0 | 0.0 |

\* Numbers are provided in a range to safeguard data confidentiality due to small cell numbers in the "Yes" category and to prevent the possibility of back calculation

†AAD stands for asthma and allergic diseases

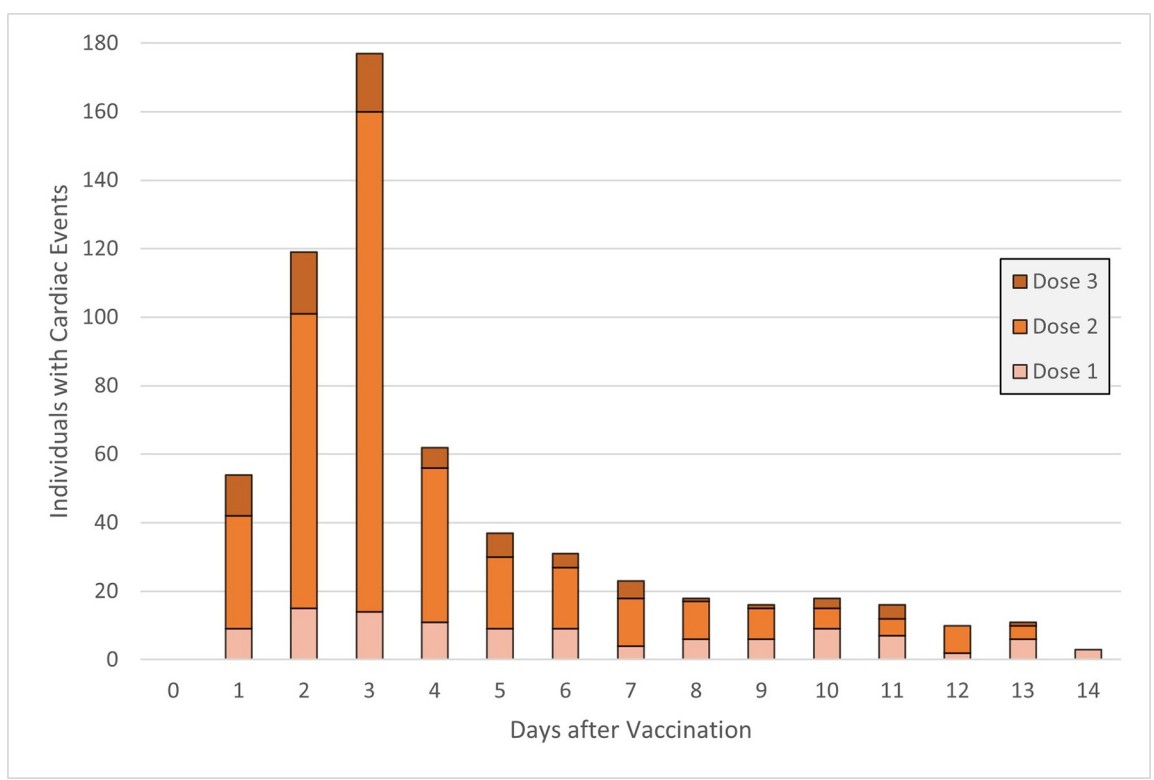

**Fig 1. Cardiac event frequency by days after vaccination.**

(OR = 2.04, 95% CI: 1.99–2.09), full (OR = 2.61, 95% CI: 2.59–2.63), and boosted (OR = 2.34, 95% CI: 2.32–2.37) vaccinations compared to non-immigrants.

## Cardiac events

The HRs, and 95% CIs of serious cardiac outcomes from multivariable negative binomial and Cox regressions are shown in Table 3. Compared to the general population's rate, the AAD population adjusted rate of serious cardiac events was significantly higher at 3.7 events per 100,000 persons (95% CI: 1.3–6.4). Adjusting for covariates, those with AAD had significantly higher associated likelihoods for a cardiac outcome compared to the healthy general population (HR = 1.31, 95% CI: 1.08–1.57). Compared to the unvaccinated, those vaccinated with original or mixed doses also had a higher associated likelihood of having a cardiac outcome within 14 days of vaccination in a similar time period (HR = 10.4, 95% CI: 4.36–25.0 and HR = 12.5, 95% CI: 7.81–20.0, respectively), adjusting for confounders. Females had a lower associated likelihood of cardiac outcomes compared to males (HR = 0.86, 95% CI: 0.79–0.93), while adolescents had higher likelihoods of cardiac outcomes compared to young adults (HR = 1.29, 95% CI: 1.07–1.54). Prior COVID-19 infection was associated with a lower probability of a cardiac event compared to non-infection (HR = 0.49, 95% CI:0.35–0.69), adjusting for covariates. S2 Table shows the breakdown of serious cardiac events by types of vaccine doses and dose number. Sensitivity analyses included stratified analyses by vaccination dose number (S3 Table), the inclusion of acute cardiac events (S4 Table). No significant differences in post-vaccine cardiac events were detected between original and bivalent doses, adjusting for confounders. Females and adolescents had significantly higher odds of cardiac events following the second dose. There were significant positive associations between diabetes and diabetes

**Table 2. Rate ratios of vaccination uptake from multivariable negative binomial regression models.**

| Covariates | Any Vaccination | | | | Full Vaccination | | | | Boosted Vaccination | | | |
|---|---|---|---|---|---|---|---|---|---|---|---|---|
| | RR* | 95% CI | | p-value | RR* | 95% CI | | p-value | RR* | 95% CI | | p-value |
| *Population* | | | | | | | | | | | | |
| AAD | 1.13 | 1.13 | 1.13 | < .0001 | 1.14 | 1.13 | 1.14 | < .0001 | 1.18 | 1.17 | 1.19 | < .0001 |
| Diabetes | 1.15 | 1.14 | 1.17 | < .0001 | 1.16 | 1.15 | 1.18 | < .0001 | 1.26 | 1.24 | 1.29 | < .0001 |
| AAD with Diabetes | 1.21 | 1.20 | 1.22 | < .0001 | 1.22 | 1.21 | 1.23 | < .0001 | 1.37 | 1.35 | 1.39 | < .0001 |
| General Population (reference) | 1.00 | | | | 1.00 | | | | 1.00 | | | |
| *Sex* | | | | | | | | | | | | |
| Female | 1.03 | 1.02 | 1.03 | < .0001 | 1.03 | 1.03 | 1.03 | < .0001 | 1.24 | 1.23 | 1.25 | < .0001 |
| Male (reference) | 1.00 | | | | 1.00 | | | | 1.00 | | | |
| *Age* | | | | | | | | | | | | |
| 12–17 | 1.12 | 1.11 | 1.12 | < .0001 | 1.12 | 1.12 | 1.12 | < .0001 | 0.71 | 0.71 | 0.71 | < .0001 |
| 18–35 (reference) | 1.00 | | | | 1.00 | | | | 1.00 | | | |
| *Residence* | | | | | | | | | | | | |
| Urban | 1.06 | 1.06 | 1.06 | < .0001 | 1.07 | 1.06 | 1.07 | < .0001 | 1.19 | 1.18 | 1.20 | < .0001 |
| Rural (reference) | 1.00 | | | | 1.00 | | | | 1.00 | | | |
| *Income Quintile* | | | | | | | | | | | | |
| 1 (Lowest) | 0.99 | 0.98 | 0.99 | < .0001 | 0.98 | 0.98 | 0.99 | < .0001 | 0.80 | 0.79 | 0.81 | < .0001 |
| 2 | 1.00 | 1.00 | 1.01 | 0.1992 | 1.00 | 1.00 | 1.01 | 0.8667 | 0.85 | 0.84 | 0.86 | < .0001 |
| 3 | 1.00 | 1.00 | 1.01 | 0.2058 | 1.00 | 1.00 | 1.00 | 0.986 | 0.88 | 0.87 | 0.88 | < .0001 |
| 4 | 1.01 | 1.01 | 1.01 | < .0001 | 1.01 | 1.00 | 1.01 | < .0001 | 0.91 | 0.90 | 0.92 | < .0001 |
| 5 (Highest, reference) | 1.00 | | | | 1.00 | | | | 1.00 | | | |
| *Recent Immigrant* | | | | | | | | | | | | |
| Yes | 1.21 | 1.20 | 1.21 | < .0001 | 1.21 | 1.21 | 1.21 | < .0001 | 1.09 | 1.08 | 1.10 | < .0001 |
| No (reference) | 1.00 | | | | 1.00 | | | | 1.00 | | | |

* Also adjusted for instability, deprivation, dependency, and ethnic diversity quintiles

†AAD stands for asthma and allergic disease

with AAD comorbidities and cardiac events after including the larger sample of acute cardiac events.

## Discussion

This Canadian population-based study followed over five million individuals aged 12–35 years and showed substantially higher rates of vaccine uptake among those with chronic conditions and a temporal association of very rare, albeit notable, myocarditis and pericarditis events from COVID-19 vaccines. While our findings are consistent with those reported in the literature [4, 12, 13], this study distinguishes itself from other COVID-19 vaccine investigations as it was population-based study with the most current information on COVID-19 vaccine use, and is one of the first studies to be able to examine and compare the bivalent and original COVID-19 vaccines.

Our study found that compared to the general population, individuals with AAD and/or diabetes had 13–37% higher rates of vaccine uptake and 1.39–2.27 times higher odds of receiving booster doses. These observations in Ontario, Canada are consistent with other studies describing vaccine uptake among those with chronic conditions. A retrospective paediatric study in Israel by Hoshen et al. noted that vaccination rates and two-dose uptake were highest amongst 61,776 children with type 1 diabetes, heart failure, obesity, and asthma [14].

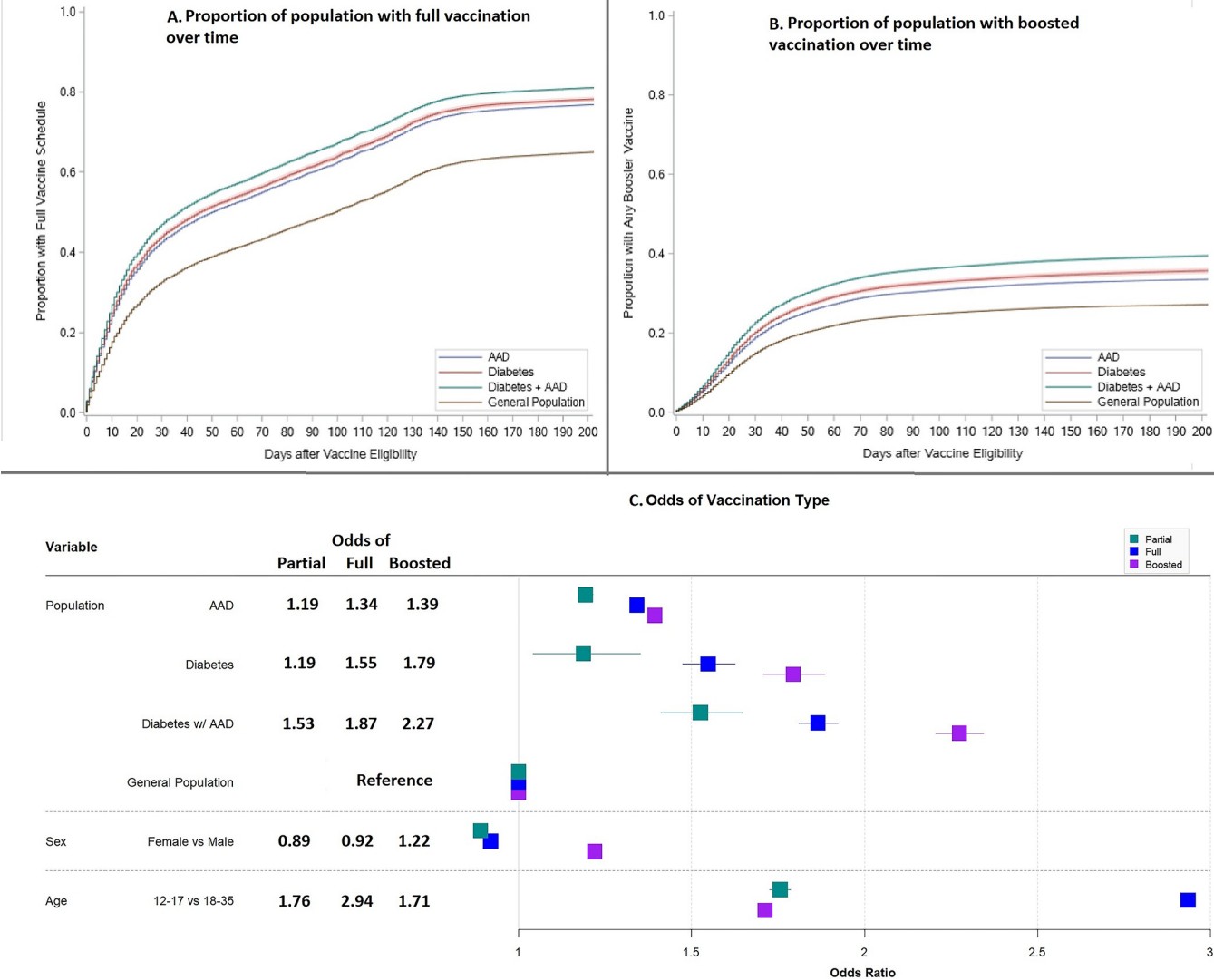

**Fig 2. Proportion of full or booster vaccination and adjusted odds of vaccination uptake.**

Bouloukaki et al.'s cross-sectional study of 626 adults in Greece also positively correlated presence of diabetes with deciding to receive a COVID-19 vaccine (r = 0.08. p = 0.043) [15]. These studies found vaccine uptake correlations early in the pandemic while our study built upon those with covariate-adjusted rate ratios.

Following the development of COVID-19 vaccines, surveillance had noted a small risk of myocarditis and pericarditis from the original vaccines. Myocarditis occurred most often among male adolescents aged 12–17 years following dose 2 [16], with median onset 2–5 days after vaccination [16], mirroring the results of our study. Our study found further evidence of these primary factors and onset time of period cardiac events. Specifically, while our rates of myocarditis/pericarditis were similar to rates reported in studies found by Fatima et al.'s systematic review [17], but different than that reported by Goddard et al. from the US Centres of Disease Control (CDC) who found lower rates after first doses (1 in 200,000 persons) and higher rates in second and first booster doses (1 in 30,000 and 1 in 50,000 persons, respectively) [18]. We also found that prior COVID-19 infection reduced the chance of myocarditis.

**Table 3. Hazard ratios of serious cardiac effects from multivariable regression models.**

| Covariates | HR* | 95% CI | | p-value |
|---|---|---|---|---|
| *Population* | | | | |
| AAD† | 1.31 | 1.08 | 1.57 | 0.0049 |
| Diabetes | 0.60 | 0.08 | 4.26 | 0.6082 |
| Diabetes with AAD† | 0.61 | 0.19 | 1.88 | 0.3859 |
| General Population (reference) | 1.00 | | | |
| *Vaccine Type* | | | | |
| Original-Bivalent Mix | 10.44 | 4.36 | 25.03 | < .0001 |
| Original | 12.50 | 7.81 | 20.01 | < .0001 |
| Unvaccinated (reference) | 1.00 | | | |
| *Prior COVID Infection* | | | | |
| Yes | 0.49 | 0.35 | 0.69 | < .0001 |
| No (reference) | 1.00 | | | |
| *Sex* | | | | |
| Female | 0.32 | 0.27 | 0.39 | < .0001 |
| Male (reference) | 1.00 | | | |
| *Age* | | | | |
| 12–17 | 1.29 | 1.07 | 1.54 | 0.0068 |
| 18–35 (reference) | 1.00 | | | |
| *Residence* | | | | |
| Urban | 0.65 | 0.49 | 0.88 | 0.0045 |
| Rural (reference) | 1.00 | | | |
| *Income Quintile* | | | | |
| 1 (Lowest) | 0.55 | 0.34 | 0.88 | 0.0131 |
| 2 | 0.67 | 0.47 | 0.96 | 0.0272 |
| 3 | 0.66 | 0.49 | 0.89 | 0.0055 |
| 4 | 0.72 | 0.56 | 0.92 | 0.0101 |
| 5 (Highest, reference) | 1.00 | | | |
| *Recent Immigrant* | | | | |
| Yes | 0.67 | 0.47 | 0.94 | 0.0207 |
| No (reference) | 1.00 | | | |

* Also adjusted for instability, deprivation, dependency and ethnic diversity quintiles

†AAD stands for asthma and allergic diseases

A prospective paediatric antibody study of 16 patients by Yonker et al. associated free spike proteins in circulation, unbound to antibodies, to vaccine-related myocarditis [19]. It is possible that COVID-19 infection prior to receiving the vaccine primed adaptive immunity responses and prevented antibody evasion by spike proteins, reducing the risk of vaccine-related myocarditis.

Furthermore, we found that individuals with AAD had a rare but higher probability of cardiac events with an adjusted rate difference of 3.7 per 100,000 persons. When expanding our definition to include acute cardiac events, we found that populations with other comorbidities (diabetes and diabetes with AAD) also had significantly higher rates of cardiac events. It is possible that the increased susceptibility from persons with allergic diseases is due to their body's propensity to hypersensitivity. Provided that most asthma patients have eosinophilic asthma, with higher eosinophil counts in the blood, they may be more sensitive to changes leading to symptoms such as difficulty breathing and inflammation, including inflammation of the heart,

also known as myocarditis [3]. Given that there are fewer serious cases among those with diabetes yet had a number of cardiac events in acute care, compared to those with AAD, it is possible that people with AAD may need more health care given their greater likelihood of respiratory symptoms.

When comparing the original and bivalent vaccines, we could not find any statistically significant difference in respect to cardiac outcomes. CDC's Hause et al.'s surveillance of 22.6 million bivalent booster doses of ages over 12 observed five vaccine-related myocarditis events [20]. Our study found a similar number of cardiac events following bivalent vaccination for the first primary and booster doses, albeit at a higher rate. It is possible that this difference is due our study's focus on adolescents and youth, which are at greater risk of cardiac events, than the adult ages included in Hause et al.'s study. This difference may also be due to the bivalent administration dose number. 95.6% of observed bivalent doses for Hause et al.'s study were for second and third boosters while 3.9% were second primary and first booster doses [20], while our study observed both primary and first booster doses, which are more likely to cause a myocarditis event.

To our knowledge, this study is the first study that examines the uptake of COVID-19 vaccines in adolescents and young adults in Canada. This study is also the first to investigate and compare the impact of monovalent and bivalent vaccines on cardiac outcomes. This study makes use of Ontario's single-payer healthcare system that allows for a very large study population with comprehensive and reliable health data records for each individual. Furthermore, our use of Ontario's health system provides us with timely information, making this study one of the first to investigate the effects of both original and bivalent vaccine doses. While we could not fully differentiate a cardiac outcome caused by COVID-19, the COVID-19 vaccine, or unrelated factors, thereby limiting our ability to find true causality, our narrow timeframe of follow-up provides strong correlative findings between cardiac outcomes and the COVID-19 vaccine.

This study also has some limitations. Despite adjusting to the best of our ability, occupation-based vaccine eligibility criteria (i.e., healthcare workers and front-line workers receiving earlier vaccine rollout dates), coupled with the lack of such information in datasets, makes it difficult to accurately determine dates of vaccine eligibility. While this affects follow-up time, we expect this to affect only a handful of individuals given the age range of our study population. Secondly, due to issues with initial vaccine rollouts, a small number of individuals went out of province/country to receive vaccine doses (<1%), skewing time to event data with lower and negative time relative to their Ontario vaccine rollout date. However, despite being out of province, many of these people are accounted for and have vaccine data available. Thirdly, COVID infection data is based on positive lab results. With advent of rapid antigen take-home tests, not everyone who caught COVID could be registered as positive, increasing the bias and likely overestimates our regression estimates of COVID on cardiac outcomes. Fourthly, this study is not designed to measure causal relations of COVID vaccination and adverse outcomes, thus our findings are therefore restricted to indicate associations. While our study's short follow-up period can imply a temporal association, our data lacked information on specific causes of myocarditis, which could be caused by other viral infections including the common cold, influenza, or COVID-19, and not necessarily due to vaccination. Lastly, the low number of serious cardiac events limited our analytical ability to detect and estimate differences between populations and vaccine types. While we could include individuals with a mix of original and at least one bivalent vaccine, we could not separate individuals who had more than one bivalent vaccine, limiting our ability to further analyse the bivalent vaccine.

## Conclusion

This Canadian population-based study demonstrated substantially higher rates of vaccine uptake among those with chronic conditions and investigated factors for a rare, albeit notable, temporally associated cardiovascular side effect of both original and bivalent COVID-19 vaccines. Our findings should further inform physicians and patients that, though it is a smaller risk than contracting COVID-19 and substantially smaller than the benefit vaccination brings, there is a continued small risk of cardiovascular events from bivalent COVID-19 vaccines especially for those with certain factors. Future research with longitudinal follow-up is needed for continued surveillance of the bivalent COVID-19 vaccines, its future vaccine iterations, and related adverse events.

## Supporting information

**S1 Appendix. Derived Ontario's COVID-19 phase 2 vaccine guideline of at-risk health conditions.**
(DOCX)

**S2 Appendix. ICD and OHIP codes for atopic diseases and cardiac outcomes.**
(DOCX)

**S1 Table. Likelihood of vaccination type from multivariable logistic regression models.**
(DOCX)

**S2 Table. Cardiac events by vaccine dose type and number.**
(DOCX)

**S3 Table. Odds of cardiac events by vaccine dose number from multivariate logistic regression models.**
(DOCX)

**S4 Table. Odds of cardiac events by vaccine dose number from multivariable logistic regression models with acute cardiac events.**
(DOCX)

## Acknowledgments

This document used data adapted from the Statistics Canada Postal Code[OM] Conversion File, which is based on data licensed from Canada Post Corporation, and/or data adapted from the Ontario Ministry of Health Postal Code Conversion File, which contains data copied under license from Canada Post Corporation and Statistics Canada. This study was supported by the Ontario Health Data Platform (OHDP), a Province of Ontario initiative to support Ontario's ongoing response to COVID-19 and its related impacts. Parts or whole of this material are based on data and/or information compiled and provided by Immigration, Refugees and Citizenship Canada (IRCC) current to May 2023. Parts of this report are based on Ontario Registrar General (ORG) information on deaths, the original source of which is ServiceOntario. Parts of this material are based on data and information compiled and provided by the Canadian Institute for Health Information, Ontario Health, and the MOH. The analyses, conclusions, opinions, and statements expressed herein are those of the authors, and not necessarily those of the data sources and are independent from the funding sources; no endorsement is intended or should be inferred.

All authors take responsibility for all aspects of the reliability and freedom from bias of the data presented and their discussed interpretation.

## Author Contributions

**Conceptualization:** Teresa To.

**Data curation:** Kimball Zhang.

**Formal analysis:** Kimball Zhang.

**Funding acquisition:** Teresa To.

**Investigation:** Kimball Zhang, Teresa To.

**Methodology:** Kimball Zhang.

**Project administration:** Kimball Zhang, Emilie Terebessy.

**Supervision:** Teresa To.

**Validation:** Jingqin Zhu.

**Visualization:** Kimball Zhang.

**Writing – original draft:** Kimball Zhang, Teresa To.

**Writing – review & editing:** Kimball Zhang, Emilie Terebessy, Jingqin Zhu, Catherine Birken, Cornelia M. Borkhoff, Andrea Gershon, Theo J. Moraes, Tetyana Kendzerska, Smita Pakhale, Teresa To.

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
