## [Decision Letter · Decision Letter 0]

13 May 2024

PGPH-D-24-00440

Uptake and Cardiac Events of COVID-19 Vaccinations Among Canadian Youth and Young Adults

Dear Dr. To,

Thank you for submitting your manuscript to PLOS Global Public Health. After careful consideration, we feel that it has merit but does not fully meet PLOS Global Public Health’s publication criteria as it currently stands. Therefore, we invite you to submit a revised version of the manuscript that addresses the points raised during the review process.

One of the reviewers has minor comments for you to address. Please try to incorporate Appendix A and B into your supplement (or have another reference handy / or delete these sentences). I note - I was able to open your supplementary tables just fine, so don't worry about that. The reviewer brings up a point about interpretation of risk ratios and odds ratios. I'm sure you just did (OR-1)*100% for the interpretation. I'm okay with that if you want to leave it in, but I also know that many readers don't learn that type of interpretation - so switching to X times higher odds using the OR directly might be better.

We look forward to receiving your revised manuscript.

Kind regards,

Abram L. Wagner, PhD, MPH

Academic Editor

Journal Requirements:

2. Please provide separate figure files in .tif or .eps format only and remove any figures embedded in your manuscript file. Please also ensure all files are under our size limit of 10MB.

4. We do not publish any copyright or trademark symbols that usually accompany proprietary names, eg ©, ®, ™ (e.g. next to drug or reagent names). Please remove all instances of trademark/copyright symbols throughout the text, including ® on page 9.

Additional Editor Comments (if provided):

Reviewers' comments:

Reviewer's Responses to Questions

**Comments to the Author**

1. Does this manuscript meet PLOS Global Public Health’s publication criteria? Is the manuscript technically sound, and do the data support the conclusions? The manuscript must describe methodologically and ethically rigorous research with conclusions that are appropriately drawn based on the data presented.

Reviewer #1: Yes

Reviewer #2: Yes

2. Has the statistical analysis been performed appropriately and rigorously?

Reviewer #1: I don't know

Reviewer #2: Yes

3. Have the authors made all data underlying the findings in their manuscript fully available (please refer to the Data Availability Statement at the start of the manuscript PDF file)?

Reviewer #1: No

Reviewer #2: Yes

4. Is the manuscript presented in an intelligible fashion and written in standard English?

Reviewer #1: Yes

Reviewer #2: Yes

5. Review Comments to the Author

Reviewer #1: The article meets Publication criteria and has relevance particularly the second objective looking at cardiac side effects of the covid 19 vaccine in young adults from Ontario, Canada. The conclusions drawn from the presented findings further supports the hypothesis of potential cardiac adverse effects of the vaccine with mentioned limitations and confounders. Minor areas of possible revision are highlighted below.

Line 11 and 217- the statistic of 39-127% higher odds of receiving booster doses. May i kindly request clarification of 127% odds of getting booster doses, does that mean more people (by a factor of 27% more) than those who got the vaccine ended up getting a booster dose or more (by 27%) people who got the covid 19 vaccine ended up getting more than 1 booster dose/or more than necessary booster doses.

Line 15,34- was asthma and ‘allergic diseases’ the only co-morbidities included in the study? this comes from the antagonistic statement stated on line 65/66 which show an exclusion of Ontario residents with chronic conditions (possibly allergic conditions as well since the criteria of exclusion is not so clear) so a clear list of the ‘allergic’ conditions included would clarify this antagonism.This list is mentioned to be available as appendix A which as a reviewer i could not find.

Line 67 and line 103- appendix A and appendix b are not available at the end of the document nor in the supplementary document provided as an .aspx file extension (on that note, the .aspx file extension is not easily opened on many devices, common pdfs and docx files could be considered).

Line 87- elaborate the acronym AAD as it is being used for the first time in this article

Line 88/89- are the examples given (allergic rhinitis and eczema) the only allergic disorders included or just examples because these 2 examples can easily be stated to be the only co-morbidities included

Reviewer #2: General Comment:

The authors did a rigorous study..... Targeting the general population for such study is commendable such that the target population of youth and young adults (or any group whatsoever; Pediatrics, the elderly, health professionals etc) health events are well spread.

The linkage and synergy of health systems data profiling is fantastic and near accurate database since data driven informatics is key for programming, not just vaccinations. Such is much desire in LLIMCs, especially our Africa.

Title: "Uptake Respiratory and Cardiac events of COVID 19 vaccinations among Canadian youth and young adults "

Keywords: COVID 19 vaccines, side effects, asthma, myocarditis, diabetes Canada

Abstract:

Line 2 - Few studies have examined population data of the COVID-19 original and bivalent vaccine data on its uptake and potential side effects

Results:

Line 147- Six hundred and sixteen (0.01%) individuals...

Line 149- Among those post-vaccinated individuals, 505(84.5%) had cardiac events within seven days and two -thirds (66.1%) of cardiac events occuring within two weeks post 2nd dose vaccination (Fig. 1)....

Table 1: The characteristics column could be such that 'The demographic factors is moved upwards then the N position as Total Population/age groups in same colored box

6. PLOS authors have the option to publish the peer review history of their article (what does this mean?). If published, this will include your full peer review and any attached files.

**Do you want your identity to be public for this peer review?** For information about this choice, including consent withdrawal, please see our Privacy Policy.

Reviewer #1: No

Reviewer #2: No

---

## [Editor Report · Decision Letter 1]

1 Jul 2024

Uptake and Cardiac Events of COVID-19 Vaccinations Among Canadian Youth and Young Adults

PGPH-D-24-00440R1

Dear Dr To,

We are pleased to inform you that your manuscript 'Uptake and Cardiac Events of COVID-19 Vaccinations Among Canadian Youth and Young Adults' has been provisionally accepted for publication in PLOS Global Public Health.

Best regards,

Abram L. Wagner, PhD, MPH

Academic Editor